# The Human Myelin Proteome and Sub-Metalloproteome Interaction Map: Relevance to Myelin-Related Neurological Diseases

**DOI:** 10.3390/brainsci12040434

**Published:** 2022-03-24

**Authors:** Christos T. Chasapis, Konstantinos Kelaidonis, Harry Ridgway, Vasso Apostolopoulos, John M. Matsoukas

**Affiliations:** 1NMR Facility, Instrumental Analysis Laboratory, School of Natural Sciences, University of Patras, 26504 Patras, Greece; 2Institute of Chemical Engineering Sciences, Foundation for Research and Technology, Hellas (FORTH/ICE-HT), 26504 Patras, Greece; 3NewDrug PC, Patras Science Park, 26504 Patras, Greece; k.kelaidonis@gmail.com; 4Institute for Sustainable Industries and Liveable Cities, Victoria University, Melbourne, VIC 3030, Australia; ridgway@vtc.net; 5AquaMem Scientific Consultants, Rodeo, NM 88056, USA; 6Institute for Health and Sport, Victoria University, Melbourne, VIC 3030, Australia; vasso.apostolopoulos@vu.edu.au; 7Immunology Program, Australian Institute for Musculoskeletal Science (AIMSS), Melbourne, VIC 3021, Australia; 8Department of Physiology and Pharmacology, Cumming School of Medicine, University of Calgary, Calgary, AB T2N 4N1, Canada

**Keywords:** myelin proteome, PPI network, metalloproteome, neurodiseases, multiple sclerosis

## Abstract

Myelin in humans is composed of about 80% lipids and 20% protein. Initially, myelin protein composition was considered low, but various recent proteome analyses have identified additional myelin proteins. Although, the myelin proteome is qualitatively and quantitatively identified through complementary proteomic approaches, the corresponding Protein–Protein Interaction (PPI) network of myelin is not yet available. In the present work, the PPI network was constructed based on available experimentally supported protein interactions of myelin in PPI databases. The network comprised 2017 PPIs between 567 myelin proteins. Interestingly, structure-based in silico analysis revealed that 20% of the myelin proteins that are interconnected in the proposed PPI network are metal-binding proteins/enzymes that construct the main sub-PPI network of myelin proteome. Finally, the PPI networks of the myelin proteome and sub-metalloproteome were analyzed ontologically to identify the biochemical processes of the myelin proteins and the interconnectivity of myelin-associated diseases in the interactomes. The presented PPI dataset could provide a useful resource to the scientific community to further our understanding of human myelin biology and serve as a basis for future studies of myelin-related neurological diseases and particular autoimmune diseases such as multiple sclerosis where myelin epitopes are implicated.

## 1. Introduction

Myelin is a multilamellar membrane that is formed by oligodendrocytes (OLs) in the central nervous system (CNS), which covers the axons. The myelinated axon can be likened to an electrical wire (the axon) with insulating material (myelin) around it. Communication between the myelin and axons is critical for axonal functions, including rapid nerve conduction, survival, and cytoskeletal organization [1]. Loss or damage to the myelin sheath results in serious neurological diseases such as multiple sclerosis (MS) [2,3,4]. In addition, myelin defects are associated with schizophrenia [5] and age-dependent decline in brain function [6].

Myelin is composed of about 40% water, and the dry mass is composed of about 80% lipids and 20% protein. Compared to other cellular membranes, myelin is unusually enriched for lipids, including cholesterol, galactolipids, and plasmalogens [7]. The dominant CNS myelin protein, proteolipid protein (PLP), displays a high affinity to cholesterol-rich membrane-microdomains [8]. The compaction of adjacent CNS myelin layers requires myelin basic protein (MBP), which attracts and compacts myelin membranes together at the major dense line [9]. MBP both displaces filamentous actin and cytoskeleton-associated proteins and saturates negative charges of the headgroups of phosphatidylinositol-4,5-bisphosphate (PIP2) on the cytoplasmic myelin membrane surfaces [10]. Additional myelin proteins have been identified, including myelin associated glycoprotein (MAG) [11], myelin oligodendrocyte glycoprotein (MOG) [12], and claudin 11 (CLDN11) [13].

Recently, the proteome of CNS myelin, purified from the brains of healthy C56BL/6N-mice, was qualitatively and quantitatively identified through complementary proteomic approaches [14]. According to this work, the most abundant myelin proteins, proteolipid protein (PLP), myelin basic protein (MBP), 2′,3′-cyclic nucleotide 3′-phosphodiesterase (CNP), and myelin oligodendrocyte glycoprotein (MOG), constitute 38%, 30%, 5%, and 1% of the total myelin protein, respectively. Additionally, the proteome profile of peripheral myelin in healthy mice and in a neuropathy model was achieved [15]. A large-scale analysis of the human myelin proteome was reported, using the shotgun approach of 1-dimensional PAGE and liquid chromatography/tandem mass spectroscopy. This proteomic approach identified proteins that have a variety of functions that are relevant for myelin biogenesis and maintenance (including signaling, cytoskeletal organization, cell adhesion, protein trafficking and vesicular transport, ion transport, endoplasmic reticulum and mitochondrial function, and energy metabolism) [16].

In this work the protein–protein interaction (PPI) network of the myelin proteome was constructed for the first time. The network comprised 2017 PPIs between 567 myelin proteins. In addition, the myelin metalloproteome was identified by a structure-based approach, and further analysis revealed that the interactions between metal-binding myelin proteins construct the main sub-PPI network of themyelin proteome. The in silico constructed PPI networks were ontologically analyzed to identify the biochemical processes and molecular functions, as well as their interconnectivity, aimed at making the myelin interactome a useful source for further studies focused on myelin-related neurological diseases. Knowledge of the myelin proteome, and in particular of myelin proteome epitopes that are recognized by T cells, is essential for the design and development of MS vaccines and therapeutics. In the last two decades, a vast number of analogues of myelin epitope peptides as MBP 83–98, MOG 35–55, and PLP 139–151 have been evaluated in human and animal models as potential immunotherapeutics against MS [2].

## 2. Materials and Methods

The myelin proteome was mined from the supplementary material of the most comprehensive myelin proteome identification study in the literature [16]. A total of 670 myelin proteins were used to construct the PPI network, and their detailed list is presented in Appendix A. 

PPIs between myelin proteins were mined from the publicly available MIntAct PPI database [17]. The metalloproteins were selected after applying a systematic bioinformatics approach used previously to identify putative metalloproteomes in various organisms [18,19,20,21,22,23,24,25]. Briefly, the approach combined strategies based on structural data and annotation for the identification of metalloproteins by searching for known metal-binding domains in their sequences. Lists of metal-binding domains were extracted from the Pfam library and 3D structures of known metalloproteins available from the PDB and MetalPDB [26]. Every proteome was analyzed for the relevant Pfam metal-binding domains with the search tool HMMER [27]. Biological function and disease enrichment analysis of the PPI networks was performed using the inBio Map™ PPI database version (2016–09) (Intomics, Kongens Lyngby, Denmark) [28]. PPI networks were visualized and analyzed using the Cytoscape software v 2.8.2 (Cytoscape Consortium, San Diego, CA, USA) [29].

## 3. Results

The PPI network of myelin comprises 2017 PPIs involving 567 proteins (Figure 1A, Table 1 and Appendix A and Appendix A). The structure of the reconstructed network is scale-free (Figure 1B). The majority of nodes (proteins) in scale-free networks have only a few connections to other nodes, whereas some nodes (hubs) are connected to many other nodes in the network. The degree distribution (number of nodes vs. number of connections of each node) is presented in the graph in Figure 1B, which clearly shows that a small number of nodes has high degree and a large number of nodes has a low degree. The average number of interactions per node is 6 (Table 1), and the maximum number of interactions in this network is 46; the 10 proteins with the highest number of interactions are listed in the Table 2.

Based on network analysis through inBio Map™ [28], the most significant gene ontology terms represented in the myelin interactome (Figure 2A) refer to the generation of precursor metabolites and energy, granulocyte activation, neutrophil degranulation, neutrophil activation involved in immune response, and ATP metabolic process (Table 3, *p* < 10^−67^). The most significant pathways associated with the nodes of the PPI network (Figure 3A) refer to vesicle mediated transport, membrane trafficking, nervous system development, neutrophil degranulation, and L1 cell adhesion molecule (L1CAM) interactions (Table 3, *p* < 10^−54^). In addition, the most significant diseases associated with the myelin proteins of interactome (Figure 4A) refer to peripheral nervous system disease, neuropathy, tauopathy, Alzheimer’s disease, and autonomic nervous system neoplasm (Table 3, *p* < 10^−54^).

Amongst the 567 proteins that constitute the myelin PPI network, 119 (~20% of the nodes) metal-binding proteins were identified (Figure 5 and Appendix A). The most significant gene ontology terms represented in the myelin PPI network between metal-binding proteins (Figure 2B) refer to the generation of precursor metabolites and energy, ATP metabolic process, energy derivation by oxidation of organic compounds, cellular respiration, and NADH metabolic process (Table 4, *p* < 10^−12^). The most significant pathways associated with the nodes of the PPI network of the metalloproteome (Figure 3B) refer to haemostasis, ion transport by P-type ATPases ion homeostasis, platelet activation, signaling and aggregation, and reduction of cytosolic Ca2+ levels (Table 4, *p* < 10^−13^). In addition, the most significant diseases associated with the interconnected myelin metal-binding proteins (Figure 4B) refer to peripheral nervous system disease, toxic encephalopathy, delta and beta-thalassemia, thalassemia minor, and autonomic nervous system neoplasm (Table 4, *p* < 10^−7^).

## 4. Discussion

Although the human and mouse myelin proteome has been identified in the past [14,15,16,30], the corresponding PPI network is not yet available. In this work, the PPI network of the myelin proteome was constructed based on available information from experimentally determined protein interactions in PPI databases. The resultant interactome is comprised of 2017 PPIs involving 567 proteins (Figure 1A). The network showed a scale-free connectivity distribution following the same behavior of most biological networks. The scale–free topology plays an essential role in the stability and rigidity of the network, conferring tolerance to random node removal but showing a high sensitivity to the targeted removal of hubs [31]. Based on the analysis of the presented PPI network of myelin proteome, highly connected proteins are essential for cell survival. As expected, several classical myelin proteins, such as 2′,3′-cyclic nucleotide 3′-phosphodiesterase (CNP), myelin associated glycoprotein (MAG), myelin basic protein (MBP), myelin oligodendrocyte glycoprotein (MOG), oligodendrocyte-specific protein (OSP), and proteolipid protein (PLP), were found in the PPI network, but there are additional proteins present in the network with higher connectivity. It is known from previous proteomic analyses that many of these are characterized as quantitatively “minor” myelin proteins; however, based on the present analysis, they may play an important role in myelin biology. Specifically, the most interconnected proteins in the PPI network are the 14-3-3 protein zeta/delta and theta (Table 2). The 14-3-3 proteins were originally identified as a family of proteins that are highly expressed in the brain. They bind to a large number of partners, usually by recognition of a phosphoserine or phosphothreonine motif [32]. The 14-3-3 protein zeta/delta is an adapter protein implicated in the regulation of a large spectrum of signaling pathways, such as glutamate receptor signaling via binding to homer homolog 3 (Homer 3) and in cytoskeletal rearrangements and spine morphogenesis by binding and regulating the activity of the signaling complex formed by G protein-coupled receptor kinase-interactor 1 (GIT1) and p21-activated kinase-interacting exchange factor beta (betaPIX) [33]. The third protein in the connectivity ranking is the heat shock protein HSP 90-alpha. Hsp90 is a major regulator of protein folding via chaperone activity in concert with other Hsps like Hsp70 [34]. Hsp90 has a major role in signal transduction by regulating signaling molecule localization, complex/scaffold formation, and acute signaling activation [35].

Among the 567 proteins that were identified in the PPI network, we detected a variety of proteins that were involved in signaling, cell adhesion, protein trafficking and vesicular transport, ion transport, energy metabolism, cytoskeletal organization, and nervous system development. All these functions are critical for myelin biogenesis and maintenance. Surprisingly absent from the PPI Network were calcium-dependent (or calcium-independent) phospholipases (PLAs), which are crucial in the demyelination and homeostasis of mammalian cell membranes [36]. Moreover, PLAs are often encountered as major components of animal venoms, particularly snake venoms. As such, they rapidly degrade myelin, resulting in localized and systemic inflammatory responses in envenomated hosts [37]. Human PLAs (specifically isoforms of PLA2) have been recently implicated in myelin degenerative diseases [38], e.g., Krabbe disease, a fatal inherited lipid disorder linked to toxic accumulation of galactosylceramidase (psychosine), which induces cell death of astrocytes and oligodendrocytes [38]. Inhibition of PLA2 can attenuate psychosine-induced astrocyte cell death. 

The presence of a large number of cytoskeletal proteins (septins 2, 5, 6, 8, 9, and 11) in the myelin PPI network proteome was not unexpected, because the cytoskeleton is a necessary component for the formation and maintenance of the myelin sheath [39]. Biogenesis and maintenance of myelin require vesicular transport for the sorting and trafficking of proteins and lipids by oligodendrocytes (OLs) to the myelin membrane. Several transport proteins were identified in the PPI network. For example, the small GTPase family of proteins, including those that are expressed in OLs (Rab3a, 5a, 5c, 6, and RalA) [40,41]; syntaxins, involved in vesicle fusion, including syntaxin-4, which is up-regulated during OL differentiation [42]; and clathrin coat-related proteins, involved in vesicle budding, including clathrin heavy chain, AP1b1, APa1, and AP2a2, were identified. The multifunctional protein CLIC4, which exists in both soluble and membrane forms and is known to be localized to several membrane systems, including the trans-Golgi network, secretory vesicles, and plasma membrane [43], was detected. NDRG1, a cytoplasmic protein that is expressed by OLs [44] was identified.

Several metal-binding proteins have been identified in the PPI network, substituting the major subnetwork in the interactome of myelin. The present study found metal binding proteins that are involved in various pathways: antioxidant activity (Superoxide dismutase (Cu-Zn), which destroys toxic radicals), metal homeostasis including ion transport by P-type ATPases (Sodium/potassium-transporting ATPase subunit alpha-1 and plasma membrane calcium-transporting ATPase 2) and ion sensors (Synaptotagmin-1, which is a calcium sensor that participates in triggering neurotransmitter release at the synapse). These findings support the importance of metals and their cell homeostasis in myelin biology. This statement agrees with the reported association of enzymes involved in metal homeostasis with rare movement disorders, for instance, sodium/potassium-transporting ATPase alpha subunits, which are identified in the PPI network, are involved in Dystonia 12 (DYT12). Dystonia 12 is an autosomal dominant dystonia-parkinsonism disorder, in which patients develop dystonia and parkinsonism between 15 and 45 years of age. It is defined by the presence of sustained involuntary muscle contractions, often leading to abnormal postures [45].

Many of the metal binding protein/enzymes interconnected in PPI network are involved in various diseases, including myelin degeneration, amyloidosis mitochondrial disorders arising from dysfunction of the mitochondrial respiratory chain, toxic encephalopathy, and thalassemia. Specifically, mutant Superoxide dismutase (Cu-Zn) accumulation induces myelin degeneration and promotes amyotrophic lateral sclerosis [46]. The copper binding major prion protein and calcium-regulated Gelsolin are associated with amyloidosis. Prion protein is required for neuronal myelin sheath maintenance [47], and Gelsolin can promote the assembly of monomers into filaments (nucleation) as well as sever filaments already formed [48]. Many metal binding proteins, which are part of the mitochondrial electron transport chain that drives oxidative phosphorylation and are involved in mitochondrial disorders, are identified in the myelin interactome, such as (i) complexes with iron sulfur binding motif (2Fe-2S) in their subunits [49]: Cytochrome b-c1 complex subunit Rieske and Cytochrome c oxidase subunit 5A; (ii) Cytochrome c oxidase subunit 2, which binds Cu as cofactor; (iii) Cytochrome c1, which is an iron-bound heme protein. Other iron-sulfur (2Fe-2S) complexes involved in mitochondrial disorders, such as NADH dehydrogenase flavoprotein 2 and NADH-ubiquinone oxidoreductase, which both catalyze electron transfer from NADH through the respiratory chain, were identified in the present myelin interactome [50]. Three metal binding enzymes that are associated with encephalopathy diseases were also identified: (i) calcium/calmodulin-dependent protein kinase type II subunit alpha, which functions autonomously after Ca2+/calmodulin-binding and autophosphorylation and is involved in sarcoplasmic reticulum Ca2+ transport in skeletal muscle, dendritic spine and synapse formation, neuronal plasticity [51], and intellectual developmental disorder, autosomal dominant 59 (MRD59) [52]; (ii) zinc-binding delta-aminolevulinic acid dehydratase (ALADH), which catalyzes an early step in the biosynthesis of tetrapyrroles. ALADH binds two molecules of 5-aminolevulinate per subunit, each at a distinct site, and catalyzes their condensation to form porphobilinogen [53]; (iii) manganese-binding glutamine synthetase (GS). GS catalyzes the ATP-dependent conversion of glutamate and ammonia to glutamine. Its role depends on tissue localization, e.g., in the brain, and it regulates the levels of toxic ammonia and converts neurotoxic glutamate to harmless glutamine [54]. Last, three hemoglobin subunits: alpha, beta, and delta, which are associated with the hemoglobinopathies, such as thalassemia, [55] were identified.

In addition, knowledge of the interacting map of myelin proteome and in particular of myelin proteins whose epitopes are recognized by T cells could assist in the design and development of immunetherapeutics in the future. An example is the development of immunoregulators (epitope peptides of MBP and MOG) for Multiple Sclerosis, which serve as platforms for the design, synthesis, and development of MS therapeutics and vaccines [56,57,58,59,60,61,62,63,64].

Myelin epitope peptides in mutated, cyclized, or conjugated form are examples of myelin peptides that serve as platforms for the design, synthesis, and development of multiple sclerosis therapeutics and vaccines. For instance, the 17 mer linear epitope peptide MBP82-98, known as Dirucotide, was investigated in preclinical and clinical studies as a potential drug for treating MS [56]. In addition, extensive studies using MOG35-55 and MBP83-99 epitope peptides alone, or mannan conjugated, resulted in novel findings, which render them potential vaccines for the immunotherapy of multiple sclerosis [30,57,58,59,60]. Currently, the focus of the ongoing research for the treatment of multiple sclerosis is the development of mannan-based vaccines [61] and mRNA vaccines [62,63,64].

## 5. Conclusions

The PPI network of myelin was reconstructed based on available information from experimentally determined protein interactions of myelin proteins. The size of the PPI network is 2017 PPIs involving 567 proteins. The presented PPI dataset could provide a useful resource to the scientific community to further our understanding of human myelin biology and the human diseases, such as multiple sclerosis, associated with myelin pathology and could be expected to serve as a basis for future studies. In particular, the PPI sub-metalloproteome will likely present numerous opportunities for the discovery and design of novel drugs specifically targeting the metal-binding domains. Interconnections of myelin proteins that are associated with neurological diseases could reveal potential protein targets for treatments by repurposing existing drugs. The myelin proteome found to be implicated in multiple sclerosis disease are myelin basic protein (MBP), myelin oligodentrocyte glycoprotein (MOC), and proteolipid protein (PLP); these are targets for multiple sclerosis treatments.

## Figures and Tables

**Figure 1 brainsci-12-00434-f001:**
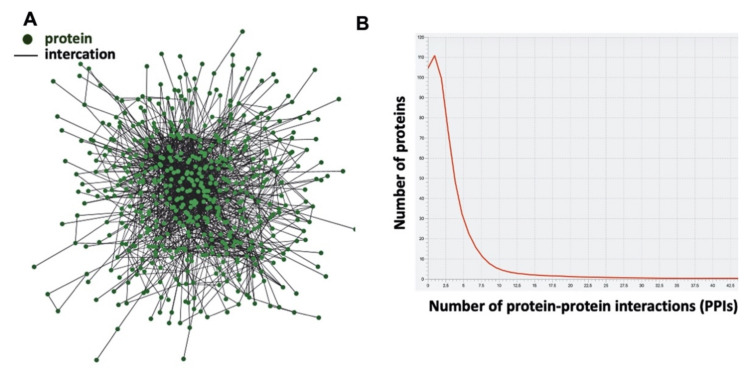
(**A**) Network graph of the protein–protein interactions (PPIs) in human myelin. Nodes represent proteins and edges, protein interactions. (**B**) The distribution of interactions for the proteins in the human myelin network.

**Figure 2 brainsci-12-00434-f002:**
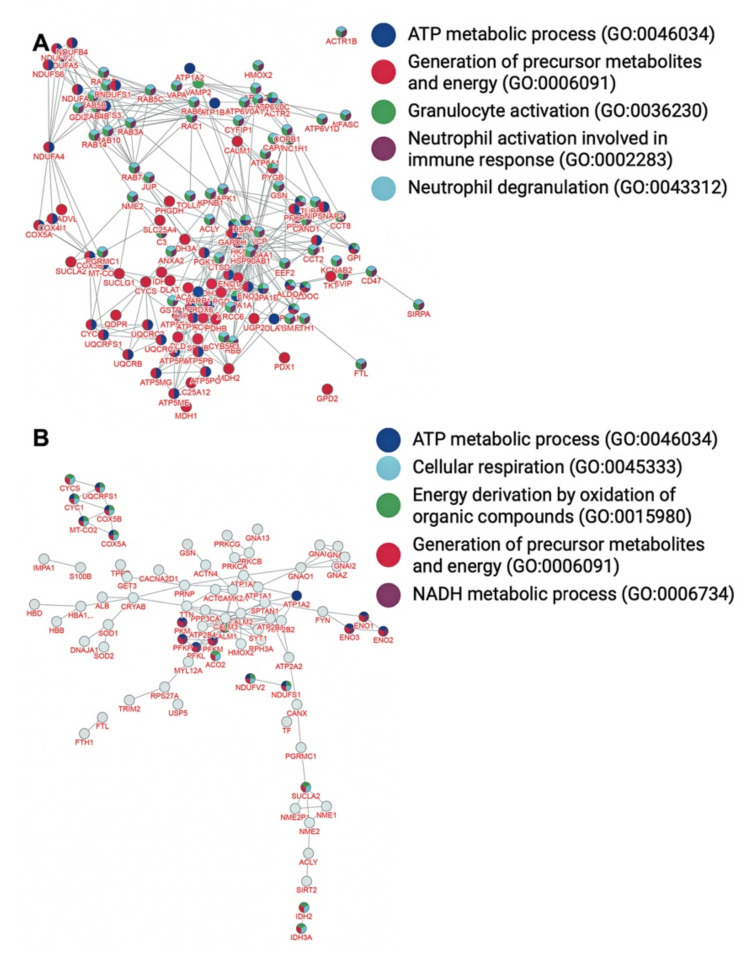
Functionally enriched PPI network of myelin (**A**) and sub-PPI network of metalloproteome (**B**) based on gene ontology (GO) terms of the proteins.

**Figure 3 brainsci-12-00434-f003:**
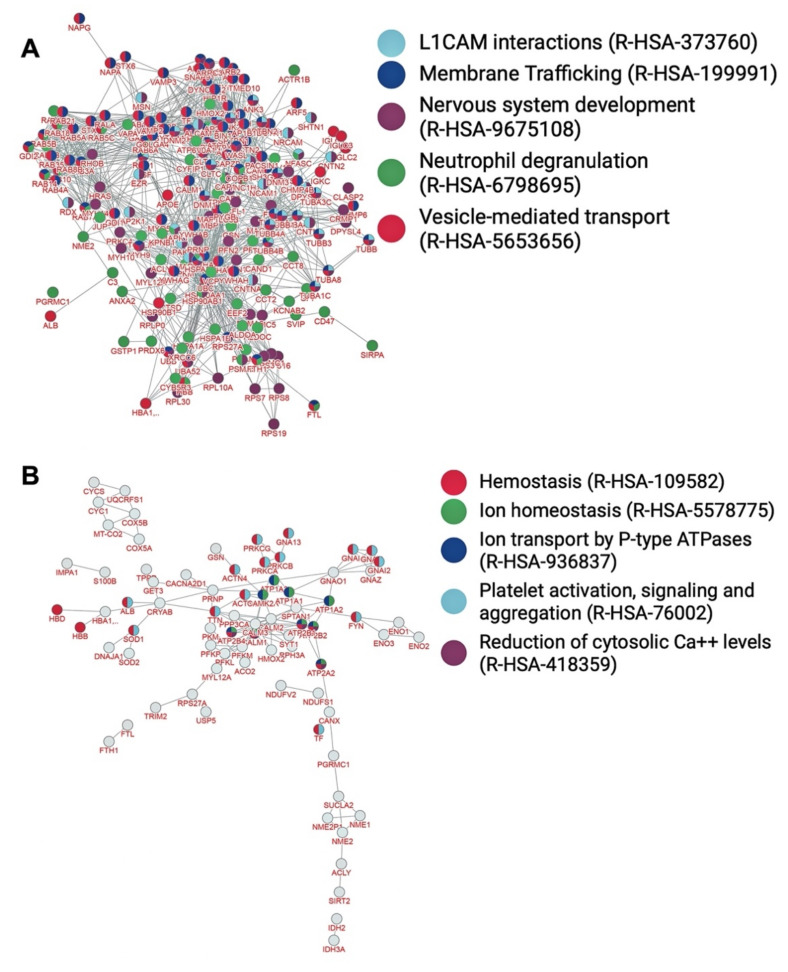
Enriched PPI network of myelin (**A**) and sub-PPI network of metalloproteome (**B**) based on most significant pathways in which myelin proteins are involved.

**Figure 4 brainsci-12-00434-f004:**
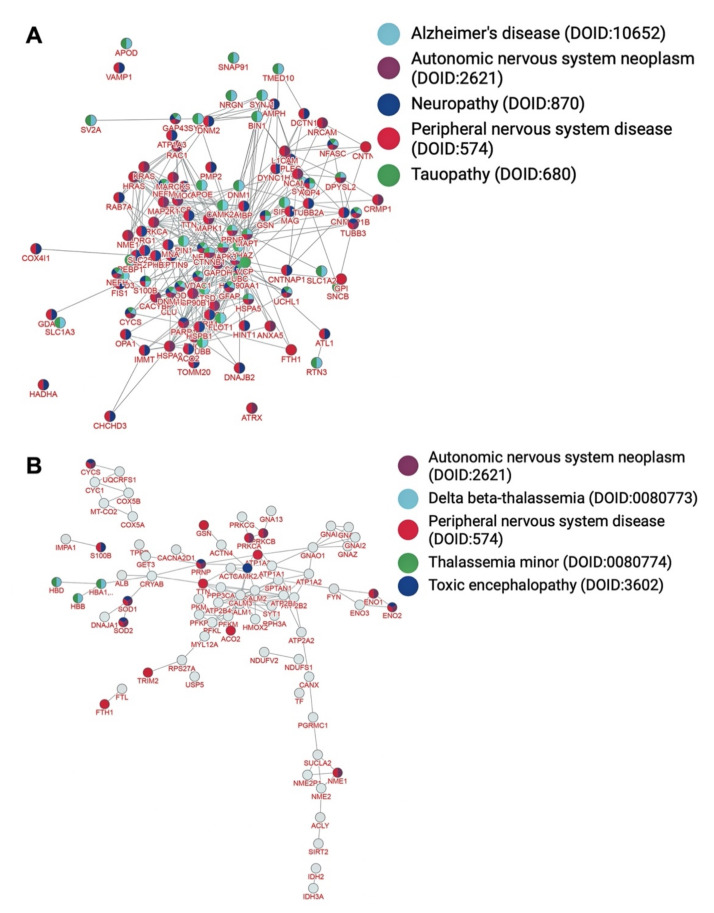
Enriched PPI network of myelin (**A**) and sub-PPI network of metalloproteome (**B**) based on associated diseases (diseasome or disease network).

**Figure 5 brainsci-12-00434-f005:**
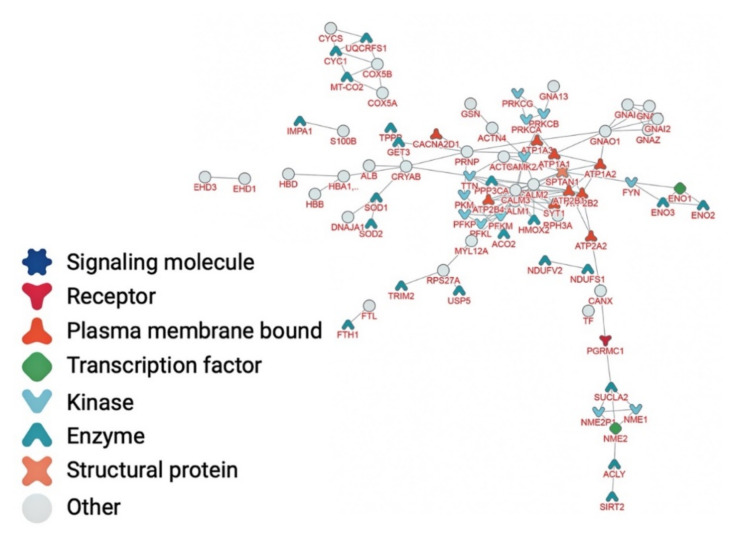
Myelin metalloproteome interacting map.

**Table 1 brainsci-12-00434-t001:** Main statistics of the myelin PPI network.

Summary Statistics	
Number of proteins	567
Number of interactions	2017
Avarage number of interactions	6.2
Network diameter	11
Number of homodimers (self-loops)	268

**Table 2 brainsci-12-00434-t002:** The top 10 proteins in the myelin PPI network that have the largest number of interactions (>24).

Entry	Gene Name	Protein Name	Degree
P63104	YWHAZ	14-3-3 protein zeta/delta	46
P27348	YWHAQ	14-3-3 protein theta	43
P07900	HSP90AA1	Heat shock protein HSP 90-alpha	41
P0CG48	UBC	Polyubiquitin-C	35
P02545	LMNA	Prelamin-A/C	32
P04792	HSPB1	Heat shock protein beta-1 (HspB1)	29
P0DP24	CALM2	Calmodulin-2	29
P17252	PRKCA	Protein kinase C alpha type (PKC-A)	28
P0DP23	CALM1	Calmodulin-1	26
P20648	ATP4A	Potassium-transporting ATPase alpha chain 1	24

**Table 3 brainsci-12-00434-t003:** Annotations of myelin protein involved in PPI network.

Biological Processes	Overlap	*p*-Value
Generation of precursor metabolites and energy (GO:0006091)	73/567	4.2 × 10^−92^
Granulocyte activation (GO:0036230)	80/567	3.9 × 10^−89^
Neutrophil degranulation (GO:0043312)	79/567	3.6 × 10^−88^
Neutrophil activation involved in immune response (GO:0002283)	79/567	4.3 × 10^−88^
TP metabolic process (GO:0046034)	49/567	2.9 × 10^−67^
**Pathways**	**Overlap**	***p*-Value**
Vesicle-mediated transport (R-HSA-5653656)	105/567	2.6 × 10^−95^
Membrane trafficking (R-HSA-199991)	95/567	4.8 × 10^−87^
Nervous system development (R-HSA-9675108)	85/567	6.6 × 10^−76^
Neutrophil degranulation (R-HSA-6798695)	79/567	4.8 × 10^−74^
L1CAM interactions (R-HSA-373760)	42/567	9.3 × 10^−54^
**Diseases**	**Overlap**	***p*-Value**
Peripheral nervous system disease (DOID:574)	91/567	<9.9 × 10^−99^
Neuropathy (DOID:870)	59/567	3.9 × 10^−69^
Tauopathy (DOID:680)	54/567	2.9 × 10^−62^
Alzheimer’s disease (DOID:10652)	53/567	9.4 × 10^−61^
Autonomic nervous system neoplasm (DOID:2621)	47/567	1.7 × 10^−54^

**Table 4 brainsci-12-00434-t004:** Annotations of myelin metal-binding proteins involved in sub-PPI network.

Biological Processes	Overlap	*p*-Value
Generation of precursor metabolites and energy (GO:0006091)	25/119	4.2 × 10^−21^
ATP metabolic process (GO:0046034)	17/119	7.7 × 10^−17^
Energy derivation by oxidation of organic compounds (GO:0015980)	17/119	2.2 × 10^−16^
Cellular respiration (GO:0045333)	15/119	6.1 × 10^−16^
NADH metabolic process (GO:0006734)	8/119	3.5 × 10^−12^
**Pathways**	**Overlap**	***p*-Value**
Hemostasis (R-HSA-109582)	30/119	2.4 × 10^−18^
Ion transport by P-type ATPases (R-HSA-936837)	12/119	3.9 × 10^−16^
Ion homeostasis (R-HSA-5578775)	11/119	1.6 × 10^−14^
Platelet activation, signaling, and aggregation (R-HSA-76002)	18/119	1.8 × 10^−14^
Reduction of cytosolic Ca++ levels (R-HSA-418359)	7/119	1.7 × 10^−13^
**Diseases**	**Overlap**	***p*-Value**
Peripheral nervous system disease (DOID:574)	20/119	8.9 × 10^−9^
Toxic encephalopathy (DOID:3602)	10/119	1.1 × 10^−7^
Delta beta-thalassemia (DOID:0080773)	3/119	2.1 × 10^−7^
Thalassemia minor (DOID:0080774)	3/119	2.1 × 10^−7^
Autonomic nervous system neoplasm (DOID:2621)	12/119	3.2 × 10^−7^

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
