# Peer review of "The Human Myelin Proteome and Sub-Metalloproteome Interaction Map: Relevance to Myelin-Related Neurological Diseases"

_brainsci, 2022, doi:10.3390/brainsci12040434_

Round 1

Reviewer 1 Report

The authors have addressed my concerns raised in the previous session. The conclusion is supported by the experimental evidence provided by the authors. Therefore, I recommend this article for publication.

Reviewer 2 Report

The manuscript has been revised according to the suggestions and comments of the reviewers.

This manuscript is a resubmission of an earlier submission. The following is a list of the peer review reports and author responses from that submission.

Round 1

Reviewer 1 Report

In this study, Chasapis et al established the protein-protein interaction network of myelin and proteome and sub-metalloproteome based on available experimentally supported protein interactions of myelin in PPI database. The authors further analyzed the gene ontology to identify the biochemical processes of the myelin proteins and interconnectivity of myelin-associated diseases in the interactomes. Overall, the authors provided a comprehensive and extensive analysis of myelin proteome and sub-metalloproteome PPI, which is very constructive and provides an useful source for further studies focus on myelin-related neurological diseases.

My main concern is that the authors emphasized that their study is particularly relevant to the multiple sclerosis therapy in their title, while they didn’t specifically provide any results or discussion that how PPI results will benefit the therapeutic development. It is important to specifically address how this study is relevant and provide specific directions, such as potential pathways or specific proteins, for therapeutic development of multiple sclerosis.

Reviewer 2 Report

The author used the supplementary data of myelin proteome, which another group published, and generated the PPI network and the sub-metalloproteome interaction map. It is an interesting project to the understanding of human myelin biology. However, I have many concerns regarding this paper.

I felt that the author did not provide sufficient explanation and information to reach the conclusion. 
The author claimed this analysis is relevant to multiple sclerosis therapy, but how protein network or classification of proteins is direct to MS therapy was not written.
"The degree distribution (number of nodes vs number of connections of each node) is presented in the graph in Figure 1B which clearly shows that a small number of nodes has a high degree and a large number of nodes with a low degree." How interpret this data? Is this pattern unique or commonly observed in proteomic data analysis? 

As the authors mentioned, myelin defects are associated with multiple sclerosis and schizophrenia and age-dependent decline in brain function, suggesting that the analysis can be used for other neurological diseases. I didn't see a reason why the authors target MS. The MS is not present in the diseases associated myelin proteomes in figure 4A. 

I felt that the introduction in lines 48-66 is not relevant information to support this study. 

The quality of figures on pdf was very poor, and I couldn't read the text in the network and graph.

References
"Additional myelin proteins have been identified, including myelin-associated glycoprotein (MAG) [11], myelin oligodendrocyte glycoprotein (MOG) [12], and claudin 11 (CLDN11) [13]." Although authors used the reference published in 2015-2018, these proteins had been identified in myelin several decades ago. The original research papers should be referred.